

# Genetic effects of long-term captive breeding on the endangered pygmy hog

Deepanwita Purohit[1], Shivakumara Manu[1], Muthuvarmadam Subramanian Ram[1,2], Shradha Sharma[1], Harika Chinchilam Patnaik[1], Parag Jyoti Deka[3,4], Goutam Narayan[5] and Govindhaswamy Umapathy[1]

[1] CSIR- Centre for Cellular and Molecular Biology, Hyderabad, India
[2] CES, Indian Institute of Science, Bangalore, India
[3] Aaranyak, Threatened Species Recovery Programme,, Beltola, Guwahati, India
[4] Durrell Wildlife Conservation Trust - Pygmy Hog Conservation Programme, Indira Nagar, Basistha, Guwahati, Assam, India
[5] EcoSystems-India, Rare & Endangered Species Conservation Unit, Beltola, Guwahati, India

Corresponding author
Govindhaswamy Umapathy,
guma@ccmb.res.in

## ABSTRACT

Long-term captive populations often accumulate genetic changes that are detrimental to their survival in the wild. Periodic genetic evaluation of captive populations is thus necessary to identify deleterious changes and minimize their impact through planned breeding. Pygmy hog (*Porcula salvania*) is an endangered species with a small population inhabiting the tall sub-Himalayan grasslands of Assam, India. A conservation breeding program of pygmy hog from six founders has produced a multi-generational captive population destined for reintroduction into the wild. However, the impact of conservation breeding on its genetic diversity remained undocumented. Here, we evaluate temporal genetic changes in 39 pygmy hogs from eight consecutive generations of a captive population using genome-wide SNPs, mitochondrial genomes, and MHC sequences, and explore the relationship between genetic diversity and reproductive success. We find that pygmy hog harbors a very low genome-wide heterozygosity ($H$) compared to other members of the Suidae family. However, within the captive population we find excess heterozygosity and a significant increase in $H$ from the wild-caught founders to the individuals in subsequent generations due to the selective pairing strategy. The MHC and mitochondrial nucleotide diversities were lower in captive generations compared to the founders with a high prevalence of low-frequency MHC haplotypes and more unique mitochondrial genomes. Further, even though no signs of genetic inbreeding were observed from the estimates of individual inbreeding coefficient F and between individuals ($F_{IS}$) in each generation, the kinship coefficient showed a slightly increasing trend in the recent generations, due to a relatively smaller non-random sample size compared to the entire captive population. Surprisingly, male pygmy hogs that had higher heterozygosity also showed lower breeding success. We briefly discuss the implications of our findings in the context of breeding management and recommend steps to minimize the genetic effects of long-term captive breeding.

## INTRODUCTION

Captive breeding has become a major conservation tool for safeguarding critically-endangered species from extinction and restoring declining populations in the wild. The primary goal of such programs is to establish a demographically-secure and genetically-healthy population through planned breeding, which will make them self-sustaining in the wild post-release (*Ballou et al., 2010*; *Ebenhard, 1995*). However, amid several successful examples of breeding and reintroduction programs (*Bolam et al., 2020*; *Price & Fa, 2004*; *Xia et al., 2014*) there is also growing evidence of reduced performance by captive-bred animals in the wild following reintroduction (*Araki et al., 2008*; *Frankham, 2008*). Factors attributed to reduced performances could be (1) inbreeding within a population with a small number of founders, (2) relaxed natural selection in captive conditions, (3) accumulation of mildly deleterious mutations and (4) adaptation to captivity (*Frankham, 2008*). Identifying these factors in a captive population is extremely difficult owing to their subtle differences in manifest effects (*Snyder et al., 1997*); however, it is essential to identify these factors in order to provide customized management solutions for an effective breeding and reintroduction program. One way of achieving this is through a comprehensive assessment of genetic diversity in captive populations through genome-wide mapping using SNPs or through a more localized approach using functional markers (*Ballou et al., 2010*). There is a growing incidence of usage of both SNPs and functional markers in population genetic studies to gain complete insight into different underlying mechanisms shaping the genetic diversity in a population (*Lenz et al., 2013*; *Yıldırım, Tolun & Tüysüz, 2011*).

The increased availability of genome-wide data of several non-model organisms over the last decade due to reduced cost of sequencing technology has enabled the study of genetic diversity patterns at a genomic scale, giving critical insights about their evolutionary processes. For example, the underlying mechanism of inbreeding depression in a small and isolated population of grey wolves in Isle Royale, Michigan, USA could be unravelled through genome-wide analysis of their heterozygosities (*Robinson et al., 2019*). Similarly, the critically endangered porpoise, vaquita, and the San Nicolas Island population of island fox showed close to zero genetic variation in their genome without losing fitness (*Morin et al., 2021*; *Robinson et al., 2016*). Furthermore, in a captive-born brown hyena, despite very low genomic diversity, there were no big stretches of homozygosity longer than or equal to 5 Mb, suggesting no conspicuous signs of inbreeding (*Westbury et al., 2018*). These empirical observations of genome-wide diversity in species of different demographic histories can guide the design and management of species-specific conservation actions.

In addition to genome-wide diversity, studies on local diversity at markers of functional significance are increasingly gaining importance since quantitative difference of some ecologically important traits is better explained by the genetic variation at functional markers (*Bonin et al., 2007*). Some of the most studied functional markers are the genes of the major histocompatibility complex (MHC), which constitute a major component of the vertebrate immune system. They code for cell surface membrane proteins that present foreign antigens to effector T cells, thereby initiating an array of adaptive immune responses (*Germain, 1994*). MHCs are highly polymorphic gene complexes that are present

in multiple copies. The high polymorphism in MHC is maintained through balancing selection mediated largely by pathogens (*Doherty & Zinkernagel, 1975*; *Hughes & Nei, 1988*; *Slade & McCallum, 1992*; *Sommer, 2005*; *Spurgin & Richardson, 2010*). Several studies have shown the direct association of MHC diversity with various fitness traits like resistance to parasites, reproductive success, and survival, owing to its role in immune responses (*Brambilla et al., 2018*; *Hedrick, 2004*; *Kalbe et al., 2009*; *Lenz et al., 2013*; *Penn, 2002*). Hence, studying MHC variability and its association with fitness in a population will inform us about the adaptive potential of a population under different environmental and demographic scenarios.

The pygmy hog is the smallest member of the *Suidae* family and the sole representative of the genus *Porcula*. Its small population size, and a very limited habitat range in the grasslands south of the Himalayan foothills puts it at high risk of extinction. A captive breeding program for pygmy hog was initiated by the Pygmy hog Conservation Breeding Program (PHCP) as a part of a species recovery program after its population plummeted to only a few hundred individuals (~150) in the wild. The program, which started in 1996 with only six wild-caught hogs from Manas National Park (MNP), Assam, has been successful in achieving its demographic goal by increasing the captive stock to a sizable number that can insure the species from extinction due to any stochastic events. With the inclusion of one rescued male in 2001 and three additional wild hogs to the captive stock in 2013 from MNP for genetic enrichment, a total of 683 individuals have been bred, comprising seven generations by 2020. The successful breeding followed by reintroduction of 130 pygmy hogs between 2008 and 2020 into their previous ranges have resulted in an increase in their numbers and range size, which prompted IUCN to downgrade the status of pygmy hog to endangered (EN) from the critically endangered (CR) category in 2019. Nevertheless, in a long-term captive rearing program, the possibility of fitness impairment due to unrecognized biases in mate-pair selection and adaptation to captivity may impact the future breeding success and compromise the reintroduction program. Therefore, the objectives of the current study are (1) to examine the genetic variability in a captive population of pygmy hogs using SNPs, mitochondrial, and MHC markers and (2) to test if the changes in genetic variability had any impact on reproductive success in the focal population. The outcomes of the present study will be helpful in devising effective management plans for breeding and reintroduction.

## MATERIALS & METHODS

### Sampling

The study sample comprising a subset of pygmy hog captive population, was collected from the Pygmy hog Conservation Breeding Program (PHCP) in Assam, India. The captive population, which has been bred up to eight generations till February 2020, is generated by breeding mate pairs based on lowest kinship and highest mate suitability index values as calculated by genealogy record keeping tool SPARKS v 1.66 and PMx. All samples were collected in accordance with the ethical and legal regulations of India and as per Wildlife Protection Act 1972. The program was approved by the Institutional
Animal Care and Use Committee and Internal Ethical committee of the Centre for Cellular and Molecular Biology vide No. IEM/CCMB 07/2020. We opportunistically collected tissue or blood samples ($n = 39$) representing all the eight generations based on their availability in the veterinary facility of PHCP. The generation and pedigree inbreeding coefficient of each individual were obtained from SPARKS v 1.66 (Table S1). The samples were composed of only two founders (PH1 & PH159) out of six. The individuals from generation 1 (PH11: progeny of a wild sire and the founder female PH4; PH21: progeny of founder male PH2 and founder female PH1; PH35: progeny of PH2 and PH4; PH41: progeny of PH2 and founder female PH5) represented the immediate progeny of the founders which were pregnant at the time of capture from the wild. As only two individuals from the founders (generation 0) could be sampled for the current study, we merged the generation 1 individuals with the generation 0 into a single group subsequently referred to as "founders". Thus, the genetic pool of five founders out of six was represented in this group. Through this step, we also ensured that the genetic pool of the wild pygmy hogs was better represented in a single group, *i.e.,* founders. Individuals from subsequent generations born through planned-breeding were considered as "captive-born". The single representatives of generations 3 and 7 were included in generations 2 and 6 respectively to ensure sample size uniformity across groups (Table S1). In SPARKS, the generation counts (T-high, T-low and T-avg) of a specimen are estimated based on the number of steps to wild ancestry. Out of the three measures, we arbitrarily chose T-high to segregate individuals into generations. Since generation 3 individuals' percentage of wild ancestry was comparable to that of generation 2 individuals and generation 7 individuals' to that of generation 6 individuals in at least one of the other measures *i.e.*, T-low and T-avg, merging of these groups was not expected to affect the pattern of temporal change in diversity. Moreover, we tested to see if the merged groups had significantly different genetic diversities.

## Fitness traits

We collected various life-history traits like lifetime breeding success (LBS), annual breeding success (ABS), juvenile survival (JS), neonatal mortality (NM), and age at first breeding for individuals which were paired for breeding in the dataset. While LBS and ABS were used as determinants of reproductive success in both males and females, JS and NM were considered only for females as piglet survival into adulthood is reliant on and is an indicator of maternal fitness (*Nowak et al., 2000*). LBS was defined as the number of offspring produced by an individual in its lifetime. The pygmy hog is a seasonal breeder and a female typically gives birth to four or five individuals per litter in each breeding season during April-June. Annual breeding success or ABS was defined as the number of offspring that survived in a given breeding season. Based on the piglets' dependence on the mother, the first two weeks after birth was considered as the neonatal period in pygmy hogs. Hence, NM was calculated as the percentage of infants, in a given litter, that died within 15 days of birth. Juvenile pygmy hogs wean at the age of three months. Therefore, JS of a female was defined as the percentage of surviving offspring up to three months in a given year. High ABS and JS, and low NM were considered as good indicators of a healthy

female. Besides these, the number of times an individual has been paired for mating in its entire lifespan was included as a covariate to check for its influence on lifetime breeding success.

## DNA extraction

Blood and tissue of piglet ear notch were collected from live animals and stored in EDTA vacutainer and 100% ethanol solution respectively. From dead animals, soft tissues of the kidney, brain, and heart were stored in 100% ethanol. All samples were stored at −20 °C until further use. The DNA was extracted using standard phenol-chloroform method (*Sambrook, Fritsch & Maniatis, 1989*). DNA purity and concentration were assessed using NanoDrop and Qubit 4 dsDNA HS assay kit respectively.

## Major histocompatibility complex

The DNA samples were used to amplify the exon 2 region of MHC class IIB gene in the pygmy hog. The primers (Forward: 5′CATTTCTTGTTTCTGGGGAAGGC3′, Reverse: 5′CCGCGGCACGAGGAAGGTC3′), designed at the intron-exon junctions of exon2, correspond to 7896 and 8195 nucleotide positions on the *Sus scrofa* chromosome 7 that harbors the *SLA-DRB1* gene. The PCR reaction yielded a product of 242 bp (excluding primers) in the pygmy hog. The amplicons were cloned in a blunt-ended cloning vector (CloneJet PCR Cloning Kit, Thermo Scientific) and transformed to *E. coli* cells of *DH5α* strain according to *Chung, Niemela & Miller (1989)*. About 20–25 isolated transformed colonies per individual were selected randomly to capture maximum variants of the MHC II gene and were subjected to Sanger sequencing in both directions. The sequences were aligned to form contigs and the diversity indices for individuals and generations were estimated. Furthermore, neutrality tests were performed to detect signs of selection. For detailed sequence and diversity analysis please refer to the Supplemental Information.

## Genome-wide SNPs
### *In silico* simulation of restriction enzyme double digestion

In order to select a suitable pair of restriction enzymes for Restriction site-Associated DNA sequencing (RADseq), we simulated the DNA fragmentation process *in silico* using DDRADSEQTOOLS (*Mora-Márquez et al., 2017*). We selected the reference genome of pig (Sscrofa11.1) as a representative sequence since the genome of pygmy hog has not been assembled yet. A set of 12 restriction enzymes (Fig. S1) containing both rare cutters and frequent cutters were selected for simulation based on the availability of a High-Fidelity (HF) version of the enzyme from the catalog of New England Biolabs (NEB, USA). We verified the in-built library of restriction sites that are packaged with DDRADSEQTOOLS with the information provided by the enzyme manufacturer and added the missing enzymes into the library. A total of 78 simulations were performed, one for each of the combinations of enzymes with replacement. The criterion for selecting the suitable pair was based on the number of DNA fragments generated between 250bp and 500bp. A threshold value of 100,000 fragments was selected which would yield a sufficient number of polymorphic sites across the genome. Finally, we selected the EcoRV and SspI enzyme combination which produced 101,967 DNA fragments, close to the desired threshold.

## ezRAD library preparation and next-generation sequencing

A total of 36 samples out of 39 passed the required purity and concentration for the preparation of high-throughput sequencing libraries. Two micrograms of genomic DNA from each sample was as digested with 20 units of EcoRV-HF and SspI-HF at 37 °C. The digestion was carried out sequentially with two hours of incubation for each enzyme, followed by a heat inactivation step at 65 °C for 10 min. For library preparation, a minimum of one microgram of double digested DNA was used as starting DNA concentration for each sample. We adapted the ezRAD protocol from *Toonen et al. (2013)* to prepare the libraries using universal Illumina adapters. We used the NEBNext Ultra II DNA library prep kit with sample purification beads for the preparation of libraries from the double-digested DNA samples. We skipped the DNA shearing by ultra-sonication step and followed the rest of the protocol according to the manufacturer's recommendations. Briefly, the double-digested DNA fragments were end-repaired, 5 prime phosphorylated, dA tailed, and ligated with the indexing adaptors from the set one, two, and three of NEBNext Multiplex Oligos for Illumina. The ligated fragments were then excised with USER (Uracil-Specific Excision Reagent) enzyme and size-selected for 500 bp using the SPRI beads. The purified adaptor-ligated fragments were enriched using PCR for 10 cycles. The enriched libraries were purified and the fragment size distribution was verified using Agilent Bioanalyzer High Sensitivity DNA chip. The libraries were quantified with Qubit 4 dsDNA assay and the concentration was normalized prior to pooling. The pooled libraries were loaded onto an entire SP flow cell and sequenced for 500 cycles on the Illumina Novaseq 6000 next-generation sequencer at CCMB.

## ezRADseq data analysis

The raw base call files from the sequencer were demultiplexed and the paired-end adapters were detected and removed using bcl2fastq v2.20 provided by Illumina (https://sapac.support.illumina.com/sequencing/sequencing_software/bcl2fastq-conversion-software.html). The raw paired-end fastq files were processed using the Stacks v2.54 pipeline (*Rochette, Rivera-Colón & Catchen, 2019*). First, the source code of Stacks was modified before compiling to include the SspI enzyme restriction sites, which is not supported in the software by default. The raw reads were cleaned and restriction site-associated DNA fragments were extracted using the *process_radtags* program. Reads with uncalled bases and low-quality scores were filtered with default parameters. The RAD-tags longer than 250 bp containing at most one mismatch in the restriction site overhang due to PCR and sequencing errors were rescued by verifying the overhangs left by double digestion of EcoRV and SspI. The extracted RAD-tags were processed using the *de novo* pipeline of Stacks for assembling the loci and genotyping. The number of allowed mismatches in a locus within and between individuals was limited to two in the *ustacks* and *cstacks* programs respectively. After assembling the RAD loci, contigs were built using the paired-end reads, and genotypes were called at 0.05 significance level with the *gstacks* program. The summary statistics such as genome-wide heterozygosity ($H$) and $F_{IS}$ were calculated using the populations program of Stacks. The $H$ for each individual was generated by considering only the number of heterozygous sites among all the genotyped sites within a given individual. Since the

calculation of individual heterozygosity is independent of the number of shared SNPs across samples, we did not employ any filtering of missing sites. The SNPs were exported into a VCF format from Stacks and further analyzed using VCFtools (*Danecek et al., 2011*). The individual inbreeding coefficient (F) and pairwise kinship coefficient ($\Phi$) were derived using the *het* and *relatedness2* flags, respectively, in VCFtools.

## Mitochondrial genome assembly and analysis

We assembled the mitochondrial genomes of different individuals using high copy randomly fragmented reads generated from the ezRADseq method (See Supplemental Information for details).

Number of unique haplotypes (h), haplotype diversity (Hd) and nucleotide diversity ($\pi$) were calculated from complete mitochondrial genomes using DnaSP v.6. The change in diversity over time was estimated by comparing the genetic variability from founders to the current breeding population. Additionally, the same indices were calculated separately for 1,150 bp of mitochondrial CytB gene and 440 bp of control region with those of *Sus scrofa*.

## Heterozygosity-fitness correlations

We used a generalized linear model framework to test the correlations between genetic diversity (MHC diversity and genome-wide heterozygosity) and various fitness measures. Fitness traits like lifetime breeding success (LBS), annual breeding success (ABS), neonatal mortality (NM), and juvenile survival (JS) were considered as response variables. They were defined as functions of genetic variables in coordination with some non-genetic explanatory variables on the basis of a biologically plausible hypothesis that best explained the response variables. For example, in LBS models, we fitted the number of times an individual has been paired for breeding (no. of breeding seasons) as a covariate to account for the unequal breeding opportunity given to individuals in the captive breeding set up. To fit the discrete data of LBS, poisson distribution with the identity link function was used. To fit the continuous data of ABS, JS and NM, a Gaussian distribution with identity link function was used. A separate analysis was carried out for each genetic parameter. A null model (only intercept) was also tested for each response variable to check for any bias in the model. We modelled the association studies separately for males and females. To rule out the presence of any confounding effects between neutral and functional markers, we tested the association between genome-wide kinship coefficient of individual pairs with their MHC allelic distance. All the models were carried out using "glm" function of lme4 package in R (*Bates et al., 2015*).

The models were ranked based on Akaike information criteria for small sample size (AICc). Each model was supported with $\Delta$AICc and the cumulative AICc weight (ranging from 0 to 1). The model with the lowest $\Delta$AICc score was considered as the best-fit model for a given response parameter. The AICc was implemented using the R package AICcmodavg (*Mazerolle & Mazerolle, 2017*).

**Table 1  Comparison of genome-wide heterozygosity ($H$) in captive generations of pygmy hog relative to the founders.**

| Generations | Size (n) | $H$ (Mean ± SE) | Comparisons | $t$-value | $P$ value |
|---|---|---|---|---|---|
| Total | 36 | 0.0006 ± 2.1E–05 | | | |
| Founders* | 6 | 0.000458 ± 7.35E–05 | | | |
| Generation 2 | 7 | 0.000635 ± 2.89E–06 | Founders*-Gen2 | −4.3913 | 0.01 |
| Generation 4 | 8 | 0.000646 ± 4.06E–05 | Founders*-Gen4 | −2.7064 | 0.021 |
| Generation 5 | 6 | 0.000645 ± 2.83E–05 | Founders*-Gen5 | −2.5925 | 0.024 |
| Generation 6 | 9 | 0.000615 ± 2.63E–05 | Founders*-Gen6 | −3.9671 | 0.005 |

# RESULTS

## Genome-wide diversity

We obtained a total of 431.67 gigabases of raw data from a single sequencing run containing 36 samples. Upon demultiplexing, an average of 52.76 million (SD 11.04M) 250 bp paired-end reads were assigned to each sample. About 3.28 million (SD 1.89M) RAD-tags containing the EcoRV and SspI overhangs were extracted from each sample. A final catalog of 417,278 unique RAD-loci (stacks) were assembled from 36 samples with a coverage of 9x (SD 2.8x, Min 3.6x, Max 13.6x). An average of 399.74 sites (SE 0.13) were genotyped per locus with a mean coverage of 8.26 samples per locus. A total of 273,122 polymorphic sites were obtained with an average of 62,373.63 SNPs per sample (SD 45,026.52).

Genome-wide heterozygosity ($H$) estimates for 36 individuals ranged from 0.0002 ± 0.00006 to 0.00083 ± 0.00001 ($H_O$ ± SE); the lowest was recorded in founder male PH159 and the highest value was estimated in a fourth-generation male, PH294 (Table S1). Further we recorded a significant increase of $H$ in captive-born individuals from all generations (Gen2 to Gen6) relative to the founders (Table 1; Fig. 1A).

Estimates of individual inbreeding coefficient (F) exhibited negative values for all individuals except PH159 (Table S1). The negative estimates of F indicate a low relatedness between alleles across loci within individuals compared to the expected value under random mating (*Wright, 1949*). We also found negative $F_{IS}$ between individuals of each captive generation, suggesting no signs of genetic inbreeding (Table S2). However, we observed low to moderate levels of genomic pairwise kinship coefficient (Φ) between individuals across generations (Fig. 1B; Table S3). The highest level of kinship was second to third-degree relationships seen between individuals from fourth to sixth generations (Fig. 1B; Table S3).

## MHC diversity

We obtained a total of 68 MHC reads that contained 35 distinct alleles from 38 pygmy hog individuals. The distinct alleles were translated into 30 unique amino acid sequences of 80 residues each, without any stop codons. Each individual carried either one, two, three, or four of these alleles which points at a classical double copy of SLA-DRB1 MHC class II gene, amplified either from homozygous or heterozygous loci. Out of 35 alleles, 31 were not shared between any two individuals. Five haplotypes were unique to the wild individuals and three were shared between wild and captive-bred pygmy hogs.
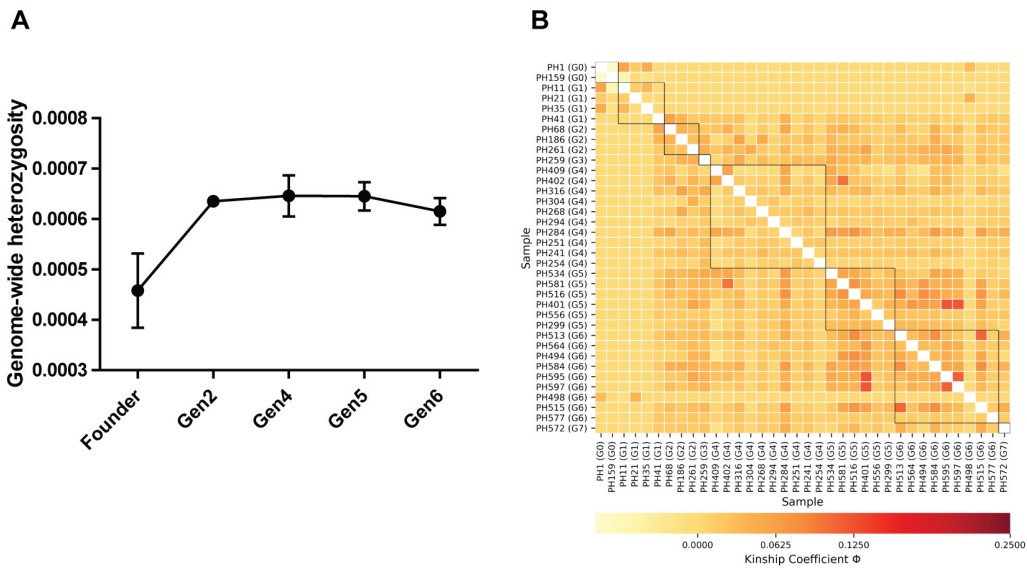

**Figure 1** (A) Comparison of genome-wide heterozygosity in pygmy hogs, (B) heatmap of kinship coefficient of 36 individuals obtained from eight generations of captive pygmy hogs.

The study population harbored a high MHC diversity with a haplotype diversity of $0.828 \pm 0.047$ (Hd $\pm$ SD) and nucleotide diversity of $0.074 \pm 0.019$ ($\pi \pm$ SD). The pattern of $\pi$ across generations showed a remarkable drop in generation 2 and 4, followed by a rise in generation 5 and 6 (Table 2; Figs. 2A & 2B). The $\pi$ value in generation 6 was slightly higher than that of the founders, showing a recovery in MHC polymorphism. The Hd showed a gradual rise in subsequent generations after an initial dip in generation 2 and 4 (Fig. 2B). Analysis of the Z test of neutral selection revealed higher rates of dS than dN along the MHC regions, although the difference between them was not significant (dN-dS $= -0.29$; $P = 0.775$). Tajima's D analysis of MHC haplotypes showed significantly negative D values in generations 2, 4, and 6, suggesting an accumulation of novel mutations (Table 2). The founders had a positive $D$ value ($D = 1.09$, $P > 0.01$). The outcome of McDonald–Kreitman test between pygmy hog and wild boar showed an $\alpha$ and neutrality index (NI) values of $-1.556$ and $2.556$ respectively, suggesting the non-synonymous to synonymous variations within populations are higher than the variations that exist between populations. However, these values were not statistically significant.

## MtDNA diversity

We assembled 22 complete and 14 partial mitochondrial genomes (mitogenomes) from the randomly fragmented reads of ezRADseq. The complete circular mitogenomes had an average length of 16,652.5 bp (SD 91.5 bp) and contained all the 13 protein-coding genes, 22 tRNA genes, and 2 rRNA genes. The partial mitogenomes had an average length of 12,086 bp (SD 3,750.8 bp) and contained scaffolded gaps filled with ambiguous bases (N). The average base coverage in the assembled mitogenomes was 159.5x (SD 110.40x,

Purohit et al. (2021), *PeerJ*, DOI 10.7717/peerj.12212

**Table 2  Summary table for MHC and Mt-DNA diversity in pygmy hogs.**

| Generations | Size (N) | MHC $\pi \pm$ SD | MHC Hd $\pm$ SD | *Tajima's D* (MHC) | Mt-DNA $\pi \pm$ SD | Mt-DNA Hd $\pm$ SD | *Tajima's D* (Mt-DNA) |
|---|---|---|---|---|---|---|---|
| Total | 35[a]/22[b] | $0.074 \pm 0.019$ | $0.828 \pm .047$ | $-2.44$ ($P < 0.01$) | $0.00022 \pm 0.00005$ | $0.775 \pm 0.08$ | $0.05681$ ($P > 0.10$) |
| Founders* | 6/6 | $0.100 \pm 0.020$ | $1.000 \pm 0.063$ | $0.83773$ ($P > 0.10$) | $0.00033 \pm 0.00006$ | $0.733 \pm 0.155$ | $1.30798$ ($P > 0.10$) |
| Generation 2* | 8/3 | $0.045 \pm 0.017$ | $0.857 \pm 0.09$ | $-1.83957$ ($P < 0.01$) | $0.00008 \pm 0.00004$ | $0.667 \pm 0.314$ | NA |
| Generation 4 | 7/7 | $0.014 \pm 0.003$ | $0.808 \pm 0.113$ | $-1.63349$ ($P < 0.05$) | $0.00007 \pm 0.00003$ | $0.714 \pm 0.181$ | $-1.43414$ ($P > 0.10$) |
| Generation 5 | 4/2 | $0.058 \pm 0.027$ | $0.818 \pm 0.119$ | $-0.86044$ ($P > 0.10$) | 0 | 0 | NA |
| Generation 6* | 10/4 | $0.106 \pm 0.053$ | $0.856 \pm 0.079$ | $-1.77101$ ($P < 0.05$) | $0.00021 \pm 0.0001$ | $0.833 \pm 0.222$ | $-0.81734$ ($P > 0.10$) |

**Notes.**

[a] Number of haplotypes for analysis of MHC diversity.

[b] Number of haplotypes for analysis of Mt-DNA diversity.

Founders*: contains wild caught pygmy hogs and their first generation progeny born outside of captive breeding program, SD represents square root of sampling variance.

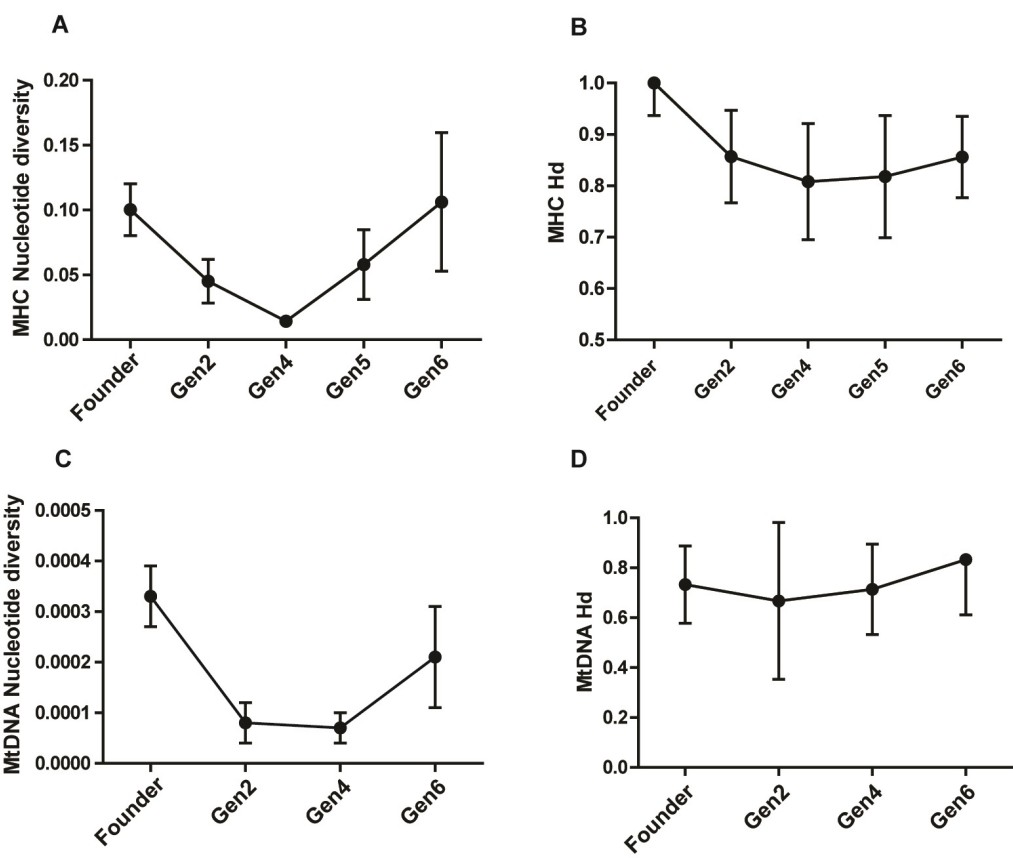

**Figure 2** **Comparison of MHC and mitochondrial DNA diversity in captive pygmy hogs (A) MHC nucleotide diversity ($\pi$), (B) MHC haplotype diversity (Hd), (C) Mt-DNA nucleotide diversity, (D) Mt-DNA haplotype diversity.** The error bars show SD or square root of sampling variance.

MIN 30.3x). For an accurate estimation of genetic variability in pygmy hogs, we sampled individuals with complete mitochondrial genomes ($n = 22$).

We found eight distinct haplotypes from 22 complete mitogenomes, of which, three were present in the founders while the remaining five were unique to the captive-bred individuals. Of the three founder-specific mitogenomes, two were shared by PH1, PH11 and PH21, and, PH159 and PH41 respectively. In spite of the shared mitochondrial haplotypes, the genomic pairwise kinship coefficient of founders ranged from $-5.93E-05$ to $0.055046$, showing low relatedness between its members. Of the 13 variable sites detected in the mitogenomes, 10 were parsimony-informative. Notably, the three singleton mutations were unique to only captive-bred individuals. The mean nucleotide diversity ($\pi$) of the entire captive population was $0.00022 \pm 0.00005$ ($\pi \pm$ SD). There was a reduction in $\pi$ in captive-born pygmy hogs compared to the founders (Table 2; Fig. 2C). The lowest estimate of $\pi$ was observed in generation 4 ($0.00007 \pm 0.00003$), consistent with the MHC results.

We also compared the diversity between pygmy hog and *Sus scrofa* in the CytB and D-loop regions in order to accomodate a larger dataset of suid sequences, including well-studied populations (Table S5). The CytB diversity in pygmy hog was low compared to

global *Sus scrofa* metapopulation as well a small population of Ryukyu wild boar (*Watanobe et al., 1999*). Similarly, D-loop diversity in pygmy hog, calculated from a 440 bp region, was lower than that of any of the 69 global *Sus scrofa* populations studied by *Zhang, Jiao & Zhao (2016)*.

## Heterozygosity-fitness correlations

We examined the correlations between individual fitness and genetic variability using three measures of diversity: genome-wide heterozygosity ($H$), MHC allelic p distance, and the number of MHC alleles per individual. In order to get an unbiased estimate of the correlation scores, we checked if the effects of $H$ and MHC diversity on fitness were mutually exclusive. Therefore, we tested if MHC p distance was associated with $H$ in the captive population. We found no correlation between these two diversity indices (linear regression: $\beta \pm SE = -6.74518 \pm 78.36237$, adjusted $R^2 = -0.03544$, $P = 0.932$; Fig. S2A. By implication, both $H$ and MHC diversity had independent effects on the focal population.

We further tested the correlation between SNP-derived pairwise kinship coefficient ($\Phi$) and MHC allelic p-distance in candidate pairs to check any signature of inbreeding in the population. We found a negative correlation between $\Phi$ and p-distance which was not significant (linear regression: $\beta \pm SE = -0.45845 \pm 0.24524$, adjusted $R^2 = 0.04193$, $P = 0.0668$; Fig. S2B). The outcome suggested that change in MHC p-distance was not affected by the degree of relatedness.

## Correlations with Lifetime Breeding Success or LBS

To check the genetic diversity on lifetime breeding success (LBS), we used an additive model. We included the number of times an individual was paired for mating (mBrseason & fBrseason) as a covariate in the interactive models to account for the higher number of offspring produced by any individual due to more breeding opportunities. We found a strong positive association between the number of times an individual mated and lifetime breeding success both in males and females. However, in the interactive model for males, $H$ was negatively correlated with LBS with high statistical significance ($-14160 \pm 6025$, $P = 0.01$; Table 3A; Fig. 3A). The high significant association with a very low residual deviance (4.99 on 8 degrees of freedom) compared to null deviance (113.8 on 10 degrees of freedom) indicated a good model fit. In other words, a larger proportion of deviance in lifetime breeding success was explained by the predictor $H$. On the contrary in females, no significant association was established between $H$ and LBS ($2876.81 \pm 4107.86$, $P = 0.48$).

In case of MHC, both its measures of diversity (number of MHC alleles per individual and MHC amino acid p distance) displayed negative correlation with LBS in male and female pygmy hogs (Table 3; Fig. 3A).

## Correlations with annual breeding success or ABS

Genome-wide heterozygosity ($H$) showed a significant negative correlation with ABS in males ($-2107.76 \pm 916.16$, $P = 0.046$) as opposed to non-significant negative correlation in females (Table 3B, Fig. 3B).

For MHC diversity, no significant associations with ABS were observed for both males and females. In females, the best ranked models established a negative correlation between

**Table 3A  Summary table for the mixed-effect models for lifetime breeding success (LBS).**

| Parameter | Lifetime breeding success or LBS, ($\beta \pm$ SE, *P* value) | $\Delta$AICc |
|---|---|---|
| Genome-wide heterozygosity ($H$) | $2876.81 \pm 4107.86$, 0.48 | 5.17 |
| | $-14160 \pm 6025$, **0.01** | 0.00 |
| Breeding seasons | $4.29 \pm 0.66$, **9.43e−11** | 0.00 |
| | $3.92 \pm 0.46$, **2e−16** | 0.76 |
| MHC allelic p distance | $-18.90 \pm 32.92$, 0.57 | 5.29 |
| | $40.34 \pm 62.31$, 0.52 | 4.27 |
| MHC alleles per individual | $-0.12 \pm 1.10$, 0.91 | 5.59 |
| | $-1.03 \pm 1.71$, 0.55 | 4.40 |

Notes.
Boxes marked in grey represent coefficient values for females.

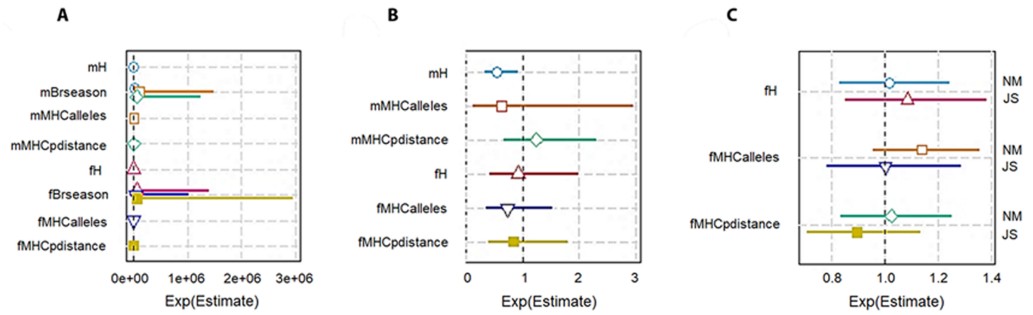

**Figure 3  Plot summary of associations between genetic diversity and different components of fitness in pygmy hogs.** H: genome-wide heterozygosity, Brseason: number of breeding seasons an individual was paired for mating, MHCalleles: number of unique MHC alleles found in an individual, MHCpdistance: mean distance between MHC alleles in an individual. Parameters prefixed with "m" are for male pygmy hogs and "f" for female pygmy hogs. (A) Lifetime breeding success or LBS model, (B) annual breeding success or ABS model, (C) neonatal mortality or NM and Juvenile survival or JS models. Bars represent the standard errors of the coefficient for each parameter. Bars not overlapping with 1 or 1e + 06 represent significant results.

**Table 3B  Summary table for the single effect models for annual breeding success (ABS), neonatal mortality (NM) and juvenile.**

| Parameter | Annual breeding success or ABS ($\beta \pm$ SE, *P* value) | $\Delta$AICc | Neonatal mortality rate or NM[*] ($\beta \pm$ SE, *P* value) | $\Delta$AICc | Juvenile survival or JS[*] ($\beta \pm$ SE, *P* value) | $\Delta$AICc |
|---|---|---|---|---|---|---|
| Genome-wide heterozygosity ($H$) | $-343.789 \pm 1704.02$, 0.8468 | 3.25 | $67.755 \pm 452.757$, 0.886 | 15.33 | $358.013 \pm 537.876$, 0.530 | 0.87 |
| | $-2107.759 \pm 916.159$, **0.0469** | 0.00 | | | - | |
| MHC allelic p distance | $-3.119 \pm 7.164$, 0.6785 | 3.31 | $0.45 \pm 1.922$, 0.8225 | 15.33 | $1.993 \pm 2.228$, 0.40554 | 0.07 |
| | $6.039 \pm 8.739$, 0.507 | 10.12 | | | | |
| No. of MHC alleles per individual | $-0.363 \pm 0.445$, 0.4465 | 0.00 | $0.157 \pm 0.107$, 0.193 | 0.00 | $0.003 \pm 0.153$, 0.985 | 0.77 |
| | $-0.453 \pm 0.784$, 0.57720 | 10.53 | | | | |

Notes.
Boxes marked in grey represent coefficient values for females.
*NM and JS parameters were modelled only for females.

ABS and MHC diversity (MHC amino acid p distance: $-3.1190 \pm 7.1638$, $P = 0.68$, $\Delta$AICc $= 3.31$, $\Sigma \, \omega i = 0.88$; number of MHC alleles per individual: $-0.3628 \pm 0.4454$, $P = 0.44$, $\Delta$AICc $= 0.00$, $\Sigma \, \omega i = 0.63$; Table 4; Fig. 3B).

### Correlations with neonatal mortality and Juvenile survival

The heterozygosity effect on neonatal mortality and juvenile survival was studied only in females. None of the diversity measures showed any significant association with NM and JS (Table 4; Fig. 3C).

## DISCUSSION

### Genetic variability in captive pygmy hogs

Maintaining genetic variability in managed populations is a key priority in species recovery programs owing to its established role in improving the adaptive potential of a population essential for successful reintroduction (*Santamaría & Mendez, 2012*). For instance, populations of chalkhill blue butterfly that had higher expected heterozygosity also showed greater lifetime expectancy (*Vandewoestijne, Schtickzelle & Baguette, 2008*). Higher success rate in species recovery programs following genetic enrichment was also observed in black-footed ferrets where individuals and their offspring produced through artificial insemination using long cryopreserved spermatozoa were integrated into their captive population (*Howard et al., 2016*). Therefore, genetic evaluation of captive populations is crucial to understanding the evolutionary impact of captive breeding on the species and to inform conservation breeding and management decisions

While our study revealed a low average genome-wide heterozygosity ($H$) in pygmy hog (Table 1 & Table S1) compared to other suids, particularly *Sus verrucosus* and European wild boar (*Liu et al., 2020*), it was comparable to Amur tiger, African cheetah and vaquita, which have historically maintained a very low population size (*Morin et al., 2021*; *Robinson et al., 2016*). The low $H$ underlines previous findings of low observed heterozygosities in pygmy hog using microsatellite markers (*Purohit et al., 2020*). Moreover, the low standard deviation of individual estimates of $H$ suggests that the paucity of heterozygous sites is evenly distributed in the sequenced regions of the genome. This pattern of low genomic variability and an even distribution of homozygosity is indicative of a small and stable population persisting over a very long time (*Morin et al., 2021*; *Robinson et al., 2019*; *Westbury et al., 2018*). This observation is supported by a recent analysis of the historical demography of pygmy hog, which estimated a low effective population size (Ne) of approximately 500 since the late Pleistocene, *i.e.,* between 100,000 to 10,000 years ago (*Liu et al., 2020*).

Interestingly, we recorded a significant increase in $H$ in captive-bred pygmy hogs relative to the founders (Table 1). In a study of genomes of captive and wild pygmy hog, *Liu et al. (2020)* observed fewer runs of homozygosity (ROH) in captive individuals. The increased $H$ as well as the shorter ROH in the captive population could have resulted from the selection of divergent individuals for mating. Prevalence of higher heterozygosity in captive populations relative to founders was previously observed in white-footed mice that were bred to minimize kinship (*Willoughby et al., 2017*). Similarly, in drosophila,
experimental lines resulting from the equalization of parental contributions retained higher heterozygosity and allelic richness compared to non-managed lines (*Rodríguez-Ramilo, Morán & Caballero, 2006*). Since the pygmy hog mate pairs were selected using similar criteria of minimized kinship between individuals and equalized contribution of founder genotypes from both the parents, we attribute the higher $H$ in captive-born individuals to the conservation- breeding protocol. This was further indicated by negative inbreeding coefficient values in the captive-bred population. Although the negative inbreeding coefficient values indicated no significant inbreeding in the captive population, the kinship coefficient showed a slightly increasing trend in the recent generations. We suspect that this pattern could possibly be due to the small sample size of non-randomly sampled individuals analysed relative to the total captive population size. Examples of such a pattern in captive populations of big mammals are very few and include a simulation study on captive Arabian oryx (*Putnam & Ivy, 2014*). Thus, our study on pygmy hog presents a key empirical evidence of heterozygosity excess in captive-bred animals.

The pygmy hog conservation-breeding program has been successful in breeding 644 individuals by 2019 which started with an initial founder stock of six individuals in 1996. This recent growth in numbers is also reflected in their genetic data through higher $H$ in the captive generations (Table 1). In addition, several singleton variable sites in the MHC regions along with unique mitogenomes were specific to the captive population, possibly due to the incorporation of novel mutations in recent times. Further, MHC regions harbored a high haplotype diversity and low nucleotide diversity at the same time which indicated that haplotypes are closely related (Table 2, Figs. 2A & 2B). This concurrence of high Hd and low Pi in a population is reflective of an expanding population after a period of low effective population size (*Grant & Bowen, 1998*; *Mendez-Harclerode et al., 2007*). Although an increase in genetic diversity is generally presumed to be beneficial, it has to be noted that mutations that are specific to captive populations can facilitate adaptation to captivity and consequently hamper the survival of reintroduced individuals in the wild environment (*Frankham, 2008*).

The MHC alleles showed a significantly high negative Tajima's D (Table 2) in all captive-bred generations, however, this need not imply positive selection in an expanding population (*Tajima, 1989*). Despite a high neutrality index score in MK test, we cannot rule out the possibility of balancing selection acting on the MHC loci due to the presence of several, novel low-frequency MHC alleles as expected for MHC loci (*Ding et al., 2021*; *Hedrick, 1998*).

The mtDNA diversity also showed a substantial drop in nucleotide diversity in captive-bred generations while the haplotype diversity was even across all the generations (Fig. 2D). In addition, both the CytB and D-loop regions showed a markedly low diversity in comparison to both the local and global populations of *Sus scrofa*. The occurrence of several low frequency mitogenome haplotypes and MHC along with the extremely low genome-wide heterozygosity and mt-DNA diversity in the captive pygmy hog population is similar to a demographic scenario where the population is in a phase of recovery following a strong bottleneck. Such populations incur heavy losses of diversity due to genetic drift and transform the intermediate-frequency alleles to rare alleles. Subsequent expansion of the
population accumulates more mutations and generates a greater number of low-frequency alleles (*Ramírez-Soriano et al., 2008*).

## Heterozygosity fitness correlations

We tested the associations of genome-wide heterozygosity ($H$) with breeding success to test if the apparent gain in genetic diversity in the captive-bred hogs had any reproductive fitness advantage. We found $H$ was negatively correlated with both lifetime breeding success (LBS) and annual breeding success (ABS) in males. In females, $H$ established a non-significant HFC with LBS and ABS. The result was unexpected as similar findings were rarely reported in animal systems. One such rare example was in a reintroduced population of Arabian oryx, which was derived from a captive breeding program after it was hunted to extinction in the wild (*Marshall & Spalton, 2000*). The authors showed that individuals with higher genomic divergence had a lower survivorship compared to individuals with intermediate divergence. *Neff (2004)* also showed a similar phenomenon in a wild population of bluegill sunfish. The authors attributed this negative association to outbreeding depression which results from disruption in pairing of locally coadapted advantageous alleles, causing reduced fitness in the progeny of two distantly related parents. The outbreeding depression in Arabian oryx could be traced back to the pairing of founders from genetically differentiated populations from its former range. The prevalence of outbreeding depression in captive pygmy hogs is debatable since the founders were picked from a single grassland range of Manas national park. However, the occurrence of negative inbreeding coefficient values for almost all individuals and the negative association between $H$ and breeding success in males, hint at the possibility of locally adapted sub-populations of pygmy hogs in the grassland range from which the founders were picked.

For MHC diversity, we found no significant correlations with any measures of fitness, both in male and female pygmy hogs (Tables 3A and 3B). However, since in comparison to $H$ the MHC analysis relied on much fewer variable sites, the power of the HFC models to resolve a significant relationship, if any, could be improved with a larger sample set.

## CONCLUSION

Genetic management of captive populations has gained importance over the last few decades and supplementing captive populations with wild individuals has been a recurrent management theme for highly endangered species (*Ballou et al., 2010*; *Lott et al., 2020*), with the genetic benefits of such an action being (1) Increasing the genetic diversity of the wild population and (2) Reducing the genetic differentiation between the wild and captive populations. On the contrary, a mandate to increase genetic diversity in captive populations may also lead to outbreeding depression if the founders belonged to genetically differentiated populations. In addition, unintentional domesticated selection resulting from a greater number of generations in captivity may affect the fitness of captive-bred individuals in the wild. Therefore, any conservation-breeding program reliant on genetic management of its captive population must also exercise caution in demographic management. Our current findings of high genome-wide heterozygosity and negative inbreeding coefficients in captive pygmy hogs have proven that long-term planned breeding of pygmy hogs has

produced a genetically heterogeneous population. Further studies on identifying unique genetic changes and their impact on individual fitness using a larger sample set will help detect adaptation to captivity and design associated breeding management practices.

The observation of negative correlation between heterozygosity and lifetime breeding success suggests that conservation breeding protocols employed to achieve higher genetic diversity in the captive population could be overlooking the possibility of outbreeding depression. In order to avoid this, some modifications in the existing protocol of mate pair selection could be explored. Furthermore, studies on the genetic structure in wild population could be undertaken to ascertain the presence of locally adapted sub-populations that might impact the efficiency of breeding programs.

## ACKNOWLEDGEMENTS

We thank Ms. Tulasi Nagabandi of the CCMB NGS facility for her help with the library preparation and sequencing. The key partners of Pygmy Hog Conservation Program are Durrell Wildlife Conservation Trust, IUCN-SSC Wild Pig Specialist Group, Assam Forest Dept., MoEF&CC Govt. of India, EcoSystems-India and Aaranyak.

### Funding

The study was funded by a Nehru Postdoctoral Fellowship (Deepanwita Purohit) from the Extramural Research Division, Council of Scientific and Industrial Research (CSIR), Govt. of India. The funders had no role in study design, data collection and analysis, decision to publish, or preparation of the manuscript.

### Grant Disclosures

The following grant information was disclosed by the authors:
Extramural Research Division, Council of Scientific and Industrial Research (CSIR), Govt. of India.

### Competing Interests

The authors declare there are no competing interests.

### Author Contributions

- Deepanwita Purohit conceived and designed the experiments, performed the experiments, analyzed the data, prepared figures and/or tables, authored or reviewed drafts of the paper, and approved the final draft.
- Shivakumara Manu and Muthuvarmadam Subramanian Ram performed the experiments, analyzed the data, prepared figures and/or tables, authored or reviewed drafts of the paper, and approved the final draft.
- Shradha Sharma and Harika Chinchilam Patnaik performed the experiments, analyzed the data, authored or reviewed drafts of the paper, and approved the final draft.
- Parag Jyoti Deka performed the experiments, authored or reviewed drafts of the paper, samples, and approved the final draft.

- Goutam Narayan conceived and designed the experiments, authored or reviewed drafts of the paper, samples, and approved the final draft.
- Govindhaswamy Umapathy conceived and designed the experiments, analyzed the data, authored or reviewed drafts of the paper, funding, chemicals, analysis tools and sequencing, and approved the final draft.

**Animal Ethics**

The following information was supplied relating to ethical approvals (i.e., approving body and any reference numbers):

Pygmy hog samples were collected within the Pygmy hog conservation breeding program in Assam India in accordance with the ethical and legal regulations of India and as per wildlife protection Act 1972.

The program was approved by the Institutional Animal Care and Use Committee and Internal Ethical committee of the Centre for Cellular and Molecular Biology vide No. IEM/CCMB 07/ 2020.

**Data Availability**

The raw reads for ezRAD sequencing are available at SRA bioproject ID PRJNA694530. The MHC and Mitogenome sequences are available at GenBank accession ID MW695410–MW695444 and MW538276–MW538297, respectively.

**Supplemental Information**

Supplemental information for this article can be found online at http://dx.doi.org/10.7717/peerj.12212#supplemental-information.

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
