# Peer review of "Genetic effects of long-term captive breeding on the endangered pygmy hog"

_PeerJ, doi:10.7717/peerj.12212_

## Round 0.1 · original submission · Major Revisions

Dear authors,

Your manuscript has been reviewed by experts. They suggest substantial changes. Please follow their recommendations as they will review your revision.

Kind regards
Michael Wink

·

Basic reporting

Authors presented novel genetic information from the captive breeding program of Pigmy hog. This study has sufficient data to draw conclusion.

Experimental design

Experimental design was enough for the study. However, data correction could have been performed as commented in details.

Validity of the findings

Though this study is reported with a high volume of data, some stretch could have been validated by resequencing.

Additional comments

The authors presented novel genetic information from the captive breeding program of the Pigmy hog.
A mixed approach was used in the genetic assessment. When authors have access to the NGS platform than the use of conventional cloning and Sanger sequencing could have been avoided for MHC heterozygosity. I do not disrespect the cloning and sanger sequencing in MHC heterozygosity assessment, but NGS is much easier (without cloning step) than the conventional (laborious cloning) method.
The mtDNA sequences provided in supplementary files are not properly aligned and contained a lot of ambiguous nucleotides. The complete mtDNA sequences are showing several needless gaps. Author may rectify the same and resubmit a final file.
The authors detected a total of eight haplotypes in mtDNA of which only three were from the founder individuals. It means five haplotypes were generated during the captive breeding program in such a short duration (over eight generations only), it is either an interesting observation or error due to low depth sequencing, which can further be rectified by resequencing only the variable stretch using Sanger sequencing method.
Your data indicates significant improvement of genetic diversity during captive breeding; however, it should have been negatively affected due to inbreeding, any possible explanation?
Conclusion sections have several citations. Phrases with citations may be sifted to the discussion section. The conclusion can be your own statements highlighting the findings without any citations.
I could not found any Supplementary method in the submission files.

Reviewer 2 ·

Basic reporting

The report is fairly clear, describing the genetic changes in a captive bred population of pigmy hogs. Some places where clarity could be improved are:

Line 80: When the text talks of no sign of inbreeding, does this mean no sign of inbreeding depression? The text talks of hyenas but the paper cited is about narwhals.

Lines 90-93: The listed possibilities are not all different. Balancing selection includes overdominance and negative frequency-dependent selection, and diversifying selection includes balancing selection.

One major result of the study is that the heterozygosity is significantly higher in all subsequent generations relative to the founders. This is shown with t-tests. However, it might not be clear from Figure 1. This is because Figure 1 shows the mean H plus or minus the standard deviation, whereas standard errors are more usual in this context. A naïve reader, seeing the overlap in what they think are error bars, might doubt the significance of the increase from the founders to later generations.

Line 404-405: In the description of the chalk-hill blue, the causality might have been reversed in the description. It is said that populations with higher diversity have higher population sizes, as if the higher genetic diversity increases fitness which then increases population size. But it would be more straightforward to explain the high diversity as being caused (as expected) by higher population size.

In lines 441 and 442 it is said that the captive breeding programme has led to higher H and also to a lower pi for the MHC. But this doesn’t make sense. If variation has gone up genome-wide one would expect it to have increased in the MHC as well.

There is a correlation between male fitness and genetic variation which goes in the unexpected direction, i.e. there is lower male fitness in the more variable animals (and this applies to both MHC and the sequence diversity in the RADseq). The biological causes of fitness variation is not entirely clear to me. It seems that the matings are controlled by the experimenters based on data including genetic variation (or at least pedigree inbreeding) in the animals. So is a high fitness male just one that has been chosen by the experimenters to be used for many crosses?

In lines 496-504 the discussion of outbreeding depression arising in a single population is rather confusing as outbreeding depression is normally defined as resulting from matings between populations. However, there is an implication in the text that the population is, in fact, subdivided, with locally-adapted subpopulations.

The conclusions in the conclusions section are all very sensible, and the recommendations are sound. However, none of these follow in any clear way from the results presented in the paper, which are somewhat mysterious and unexplained.

Experimental design

Line 264: Here for “lifetime breeding success” a logit link function is used in the generalized linear model. But a logit link function is best suited to a variable with a defined maximum as well as a defined minimum value (as with a proportion, for example). Lifetime breeding success does not have a defined maximum value. (I can see the purpose of the Poisson.)

The standard errors in lines 289-290 seem odd. The lowest value has a SE that is 30% of the point estimate, but the highest value has a SE that is less than 2% of the point estimate. I am surprised that the SEs are so different.

There are consistently negative estimates of the genomic inbreeding coefficient. My understanding is that this means that the two haploid genomes within an individual are more diverged than are random haploid genomes from the population. This is surprising. Does this reflect a breeding policy where the individuals crossed are more diverged genetically than are random individuals? There seems to be some talk of this in lines 431-433. Or perhaps this has some artefactual cause? This is said to be reflected in the low levels of pedigree inbreeding seen. But the pedigree inbreeding seems quite high in the later generations.

In Table 2, the mean value for pi for the MHC seems much too low given the mean values given for the various generations.

Lines 360-363 say that there is no inbreeding in the captive population. This cannot be right. The pedigree inbreeding measures showed that there is certainly inbreeding. What the data in lines 360-363 show is that the measured pi values are not correlated with the degree of inbreeding.

The negative relationship between Annual Breeding Success and heterozygosity is only just significant (p=0.046), and this is just one of the twelve significance tests reported in Table 3B, and there may be a Type I error arising through multiple testing here.

Validity of the findings

See above

Additional comments

Minor typographic errors:
Line 39: “a” before “captive”
Line 44: “lower” than what
Line 55: hyphen after “demographically”
Line 58: “is” for “are”
Line 74: Add comma after “scale”
Line 84: Add “the” before “major”
Line 91: Hyphen in wrong place
Line 94: Comma after “survival
Line 99: “The pigmy hog“ for “Pigmy hog”
Line 100: Add “program” after “breeding”
Line 104: “, which” for “that”
Line 112: Add “the” before “critically”
Line 141 (and 142): “individuals’” for “individual’s”
Line 153: “The pigmy hog“ for “Pigmy hog”
Line 181: “were subjected” for “subject”
Line 200: Add comma before “close”
Line 250: Add “and” before “nucleotide”
Line 271: “individual” for “individuals”
Line 290: Remove comma after “male”
Line 402: Add “the” before “chalk-hill”
Line 518: “tends” for “tend”
Line 533: Add “the” before “wild”

---

## Round 0.2 · Minor Revisions

Dear authors,

Your manuscript requires another round of revision.

Kind regards
Michael Wink
AE

·

Basic reporting

no comment

Experimental design

no comment

Validity of the findings

no comment

Additional comments

my suggestions are fulfiilled.

Reviewer 2 ·

Basic reporting

I have read the revised version of the paper, and my initial feeling is that, while very major changes have been made to the discussion section, elsewhere there was a need for a more thorough revision of the methods and results sections. In particular, the methods section should say more about the methodology for choosing animals to be parents. I felt that I found out more about this from the authors’ rebuttal letter than I did from the paper itself.
Having read this revised version of the manuscript I felt that the authors should still add some more explanation about the relationship between the measure of H, or heterozygosity and the diversity of the population. While this was not entirely clear to me from the first version, the individual measures of heterozygosity, H, are higher than expected from random mating as a result of the breeders choosing the most diverged animals to use as parents. This is, I think, the reason why H is significantly higher in subsequent generations than it is in the founders. I can see that choosing the most diverged individuals to be parents is a good strategy for maintaining diversity. But this strategy means that H is no longer a good measure of the diversity that exists in the population. It is certainly not equivalent to pi or the gene diversity, which is the divergence between randomly chosen alleles. There are sentences such as on line 475 (in the “track-changes” numbering) that imply that H is a measure of the diversity in the population.
I wasn’t entirely convinced by the authors’ argument that there was no type 1 error expected with the significant negative correlation between the H value in males and their annual reproductive success, although I now see that the ABS values essentially restate the results for LBS, once the effect of the number of breeding attempts on LBS is taken out in the analysis.

Experimental design

Specific points about this version (numbers refer to the “track-changes” numbering):
Line 51: The paper says “we find no evidence of inbreeding”. I had questioned whether the authors meant that they had found no evidence of inbreeding depression, but they said that it was inbreeding itself that they had found no evidence for. This statement would confuse readers, as there has clearly been inbreeding going on, the pedigree shows that relatives are mating. I think that the authors mean that they have failed to find a sign of inbreeding in some specific molecular test, such as, maybe, looking for runs of homozygosity. If so, they should be more specific about this, as the population is unquestionably inbred. This is also relevant to line 95.
In line 328, I don’t think that it is true that the negative F values suggest that there is no loss of heterozygosity in the captive population. It’s the H values that show that there is no loss of heterozygosity. The negative F values suggest a possible cause for part of this effect (see above).
There is a problem with Table 2. The founders are said to have Hd of 1.00 for the MHC. This is the theoretical maximum value, showing that all MHC haplotypes were different. But if this is the maximum value, I do not see how it can have a standard error or standard deviation. The Figures 2(b) and 2(d) might be improved if the scale of the Hd on the Y axis was only shown up to a maximum of 1.00, as this is the theoretical maximum.
In lines 446-448, in my first comments I was saying that the correlation between genetic diversity and population size in the chalk hill blue was probably caused by larger populations harbouring more genetic diversity due to the reduced effects of genetic drift, rather than the authors suggestion that the genetic diversity was causing the populations to be larger. The authors have re-written this section but they still seem to be saying that the high diversity is causing the population to be larger.
In line 474 it is said that there is an increase in H in captive populations, and it is said that the observation that there are fewer runs of homozygosity is a “similar observation”. The two observations are not similar in themselves, but rather are both what might be expected if genetic diverged individuals were chosen as parents for the matings, as has happened here.

Validity of the findings

Minor Points:
Line 110: I had not seen the word “polymorphicity” before. Is the meaning of this term generally known?
Line 126: Add comma after ”program”
Line 134: Perhaps say ”downgrade” for “upgrade”
Line 329: When it says that there is low relatedness between loci, I think that this should be “low relatedness between alleles”.
Line 357: In describing the MK test, which I think is the McDonald-Kreitman test, it is said that there is no significant difference between pigmy hog and Sus scrofa, which doesn’t express clearly what an MK test is looking for, and, indeed, is quite unclear in its meaning. In a McDonald-Kreitman test the null hypothesis is that the ratio of non-synonymous to synonymous base changes in the coding sequence is the same for the variations within and between species.

---

## Round 0.3 · accepted · Accept

Dear authors

Thank you for revising your ms according to the recommendations of the reviewers. Now, your manuscript can be accepted.

Greetings
Michael Wink
Academic Editor